# Innovative Approach to Isolate and Characterize Glioblastoma Circulating Tumor Cells and Correlation with Tumor Mutational Status

**DOI:** 10.3390/ijms241210147

**Published:** 2023-06-14

**Authors:** Francesca Lessi, Mariangela Morelli, Sara Franceschi, Paolo Aretini, Michele Menicagli, Andrea Marranci, Francesco Pasqualetti, Carlo Gambacciani, Francesco Pieri, Gianluca Grimod, Vanna Zucchi, Samanta Cupini, Anna Luisa Di Stefano, Orazio Santo Santonocito, Chiara Maria Mazzanti

**Affiliations:** 1Section of Genomics and Transcriptomics, Fondazione Pisana per la Scienza, San Giuliano Terme, 56017 Pisa, Italy; m.morelli@fpscience.it (M.M.); s.franceschi@fpscience.it (S.F.); p.aretini@fpscience.it (P.A.); m.menicagli@fpscience.it (M.M.); a.marranci@fpscience.it (A.M.); c.mazzanti@fpscience.it (C.M.M.); 2Department of Radiation Oncology, Azienda Ospedaliera Universitaria Pisana, University of Pisa, 56122 Pisa, Italy; francep24@hotmail.com; 3Division of Neurosurgery, Spedali Riuniti di Livorno—USL Toscana Nord-Ovest, 57124 Livorno, Italy; carlo.gambacciani@uslnordovest.toscana.it (C.G.); francesco.pieri@uslnordovest.toscana.it (F.P.); gianluca.grimod@uslnordovest.toscana.it (G.G.); annaluisa.distefano@uslnordovest.toscana.it (A.L.D.S.); orazio.santonocito@uslnordovest.toscana.it (O.S.S.); 4Division of Pathology, Spedali Riuniti di Livorno—USL Toscana Nord-Ovest, 57124 Livorno, Italy; vanna.zucchi@uslnordovest.toscana.it; 5Division of Oncology, Spedali Riuniti di Livorno—USL Toscana Nord-Ovest, 57124 Livorno, Italy; samanta.cupini@uslnordovest.toscana.it; 6Neurology Department, Foch Hospital, 92150 Suresnes, France

**Keywords:** glioblastoma, circulating tumor cells, DEPArray, whole exome sequencing, *TERT* promoter mutations

## Abstract

Circulating tumor cells (CTCs) are one of the most important causes of tumor recurrence and distant metastases. Glioblastoma (GBM) has been considered restricted to the brain for many years. Nevertheless, in the past years, several pieces of evidence indicate that hematogenous dissemination is a reality, and this is also in the caseof GBM. Our aim was to optimize CTCs’ detection in GBM and define the genetic background of single CTCs compared to the primary GBM tumor and its recurrence to demonstrate that CTCs are indeed derived from the parental tumor. We collected blood samples from a recurrent IDH wt GBM patient. We genotyped the parental recurrent tumor tissue and the respective primary GBM tissue. CTCs were analyzed using the DEPArray system. CTCs Copy Number Alterations (CNAs) and sequencing analyses were performed to compare CTCs’ genetic background with the same patient’s primary and recurrent GBM tissues. We identified 210 common mutations in the primary and recurrent tumors. Among these, three somatic high-frequency mutations (in *PRKCB*, *TBX1*, and *COG5* genes) were selected to investigate their presence in CTCs. Almost all sorted CTCs (9/13) had at least one of the mutations tested. The presence of *TERT* promoter mutations was also investigated and C228T variation was found in parental tumors and CTCs (C228T heterozygous and homozygous, respectively). We were able to isolate and genotype CTCs from a patient with GBM. We found common mutations but also exclusive molecular characteristics.

## 1. Introduction

Circulating tumor cells (CTCs) are cancer cells that detach from the main tumor mass and enter the circulatory stream [1]. In solid tumors, they are generally associated with the appearance of metastases [2]. Nowadays, in many tumors, the role as an independent prognostic factor for CTCs is now widely demonstrated, but their clinical utility is still being evaluated [3]. Several general oncology clinical trials have been conducted and are still ongoing to translate the use of CTCs into clinical practice [4,5,6,7].

Glioblastoma (GBM) is the most aggressive and deadly primary tumor of the central nervous system (CNS) in adults [8]. Complete surgical resection of this tumor is extremely difficult because of the capacity of the GBM to infiltrate the CNS parenchyma. Inexorable intracranial recurrences occur, resulting in a dramatic worsening of the prognosis; indeed, the average survival of GBM patients is only 16 months [8]. First-line treatments, including surgery followed by concomitant radiochemotherapy and adjuvant chemotherapy, fail to achieve long-term disease control. Therefore, there is an urgent need for new targets for oncological therapies and non-invasive markers for disease monitoring in GBM patients.

Until now, CTCs have not been studied much in brain tumors because GBM only rarely gives extracranial metastases; this is partially due to the limited survival of these patients not allowing time for micro-metastases to grow and be identified in other organs [9,10]. GBM cells have tropism for brain tissue because, unlike other organs, the poor connective stroma tissue of the CNS represents a favorable environment for growth [11]. Furthermore, in 2014 Jimsheleishvili et al. [12] demonstrated that multiple solid organ transplant recipients developed extracranial GBM after receiving organs from GBM patients. Despite these theories, it is now well known that CTCs are present in the blood of patients with GBM. CTCs have been detected in the blood of patients with GBM in some studies [13,14,15,16,17] however, detection of CTCs in GBM patients is technically challenging because isolation methods are not standardized, and universal markers specific to CTCs derived from brain tumors are lacking [18]. In 2014, the first three published papers were on several CTCs isolation techniques for GBM, giving heterogeneous and divergent results [14,16,19] with generally low CTCs yields. Subsequently, other studies have demonstrated the presence of CTCs in GBM patients [13,15,17,20].

In this study, we optimized and tested a novel approach to isolate CTCs in the blood of a patient suffering from recurrent GBM IDH wild-type (wt), consisting of a first size-based enrichment of cells followed by sorting with DEPArray NxT. Blood was collected at the time of the second surgery for recurrence. Molecular investigations were performed on CTCs collected at recurrence and in recurrent tumors of origin to assess common genetic somatic alterations. We also analyzed the primary GBM tissue (FFPE) available from archives and compared genetic somatic alterations. Common somatic mutations between the primary and the respective recurrences were then sought in the CTCs (*BRAF* V600E, *TERT* C228T, and C250T), to confirm the origin of these circulating cells from the parental GBM tumor.

## 2. Results

### 2.1. CTCs Description

In our sample, we identified viable CTCs using multiparametric fluorescence analysis. Our CTC signature was based on positivity for at least one of the following markers: GFAP, EGFR, or Ki67. The CTCs observed were: 18 CTCs (tumor astrocytes, GFAP+ and/or EGFR-Ki67+), 86 nucleated hematopoietic (nH) cells (non-tumoral cells, CD45+), 12 double-positive cells (dposCTCs) (GFAP+/CD45+ or EGFR-Ki67+/CD45+), and 158 unstained cells (only Hoechst+). Cells labelled only with Hoechst were considered because their shapes and sizes have the typical characteristics of CTCs. Therefore, as there is no universal marker for GBM tumor cells, we also evaluated these cells and termed them “putative CTCs” (pCTCs). Subsequently, we sorted some CTCs for further analysis. In particular: six single CTCs, two single pCTCs, one group of pCTCs, two single dposCTCs, and two groups of nH cells. All the sorted cells were subjected to DNA amplification and low-pass analysis. Figure 1 shows examples of CTCs isolated from the blood of a patient.

### 2.2. CNA Analysis on Whole Tissues and CTCs

CNA analysis was carried out on bulk primary and recurrent tumor tissues and on CTCs as a first analysis to provide an overview of chromosomal structural abnormalities and genomic instability typical of tumoral status. Chromosomal alterations of the bulk tumor, both for the primary GBM and recurrent tumor, were calculated using the CNApp tool and are shown in Figure 2 (right side). Chromosomal amplification and deletion are shown in red and blue, respectively. The intensities of the red- and blue-colored components correlate with the gain and loss values based on the results obtained from the CNApp tool. CNA analysis revealed considerable aneuploidy with whole-arm and whole-chromosome alterations (gains and losses), demonstrating chromosomal instability. Common alterations between the primary GBM tumor and its recurrence are chromosome 9p and 10 deletions and the gain of chromosome 7p. However, the primary tumor had exclusive alterations compared to its recurrence, in particular the deletion of chromosomes 13q and 18 and the gain of chromosomes 7q and 19p.

Next, we performed CNA analysis of CTCs isolated from the patient’s blood collected at the time of recurrence and compared the results with those obtained from whole tissues. In Figure 2, a comparison between primary, recurrent tumor, and CTCs CNAs obtained with CNApp is shown. The results obtained are very heterogeneous; in general, there are a higher number of alterations in CTCs than in bulk tumor tissues. In particular, CTC#2 was the only one carrying an alteration present in the primary and recurrent tumors, that is, the gain of chromosome 7. Otherwise, several new alterations are observed in CTCs, some of which are shared, such as the deletion of chromosome 19, which is present in all CTCs (CTC#1, CTC#2, CTC#5, and CTC#6). All pCTCs present deletions on chromosome 19, except for pCTC#3, which conserves the q arm of the chromosome. Deletion of chromosome 20p was also observed in several samples (CTC#1, CTC#2, CTC#5, pCTC#2, and dposCTC#2), and deletion of chromosome 22q was observed in CTC#1, CTC#2, CTC#5, pCTC#3, and dposCTC#2.

### 2.3. Whole Exome Sequencing

Whole-exome sequencing analysis was performed on DNA obtained from FFPE tissue (primary tumor), fresh tissue (recurrence), and whole blood of the patient. The tumor DNA was analyzed using deep whole exome analysis to obtain even rare somatic variants with very low allele frequencies. In contrast, whole blood DNA was subjected to regularly covered exome analysis to obtain the signature of the germline variants of the tissue. For the primary tumor, 132 million reads were mapped to the coding regions and 141 million were mapped for recurrence. The average coverage was 224X for the primary tumor and 247X for recurrence. We discovered a total of 1629 and 1237 variants in the primary tumor and recurrence, respectively. The mutational spectrum is shown in Appendix A. Subsequently, we queried the IntOGene framework as is shown in Appendix A. By comparing the somatic mutations discovered in the primary tumor and recurrence, we identified 210 common variants. In Appendix A the mutational spectrum of the shared mutations is shown. To establish the clinical significance of the 210 shared mutations, we used VarSome [21], a search engine for human genomic variations, as shown in Appendix A. The molecular variation detected in the *BRAF* gene corresponds to the well-known *BRAF* V600E mutation, which is present in GBM at a frequency of 1%. In the primary tumor, *BRAF* V600E was present with a tumor frequency of 12%, whereas in the recurrent tumor, it was <1%, despite the fact that the patient had not been treated with *BRAF* targeted agent.

### 2.4. Detection of Patient-Tailored Tumor Somatic Variations in CTCs and BRAF V600E

Of the 210 shared mutations between primary and recurrent tissues, three mutations, on the basis of their high frequency, were chosen for investigation in CTCs. The mutations were *PKRCB* (c.430G > A), *TBX1* (c.815C > T), and *COG5* (c.1554T > G), with frequencies of 39%, 26%, and 10% in primary tumor and 25%, 19%, and 13% in recurrence, respectively. At least one of the three mutations was present in 4/6 single CTCs and all three pCTCs (single and grouped). Among the dposCTCs, only one had one of three mutations. The groups of nH cells contained no mutations. All identified mutations were present in heterozygosis. Figure 3 shows the results of the presence of the selected mutations in the CTCs together with the CNA data derived from the low-pass analysis. Since the primary and recurrent tumor tissues showed *BRAF* V600E mutation, we examined the presence of *BRAF* V600E in CTCs. All CTCs were wild type for *BRAF* V600E.

### 2.5. Detection of TERT Promoter Variations in CTCs

*TERT* promoter mutations are frequently observed in gliomas with wild-type IDH. We investigated in CTCs and in the primary and recurrent tissues, two well-known mutations of the *TERT* promoter gene (C228T and C250T). They map −124 and −146 bp, respectively, upstream of the *TERT* ATG site, respectively. In the primary tumor and recurrence, we identified a C228T mutation in heterozygosis and confirmed its presence in two CTCs, CTC#6 and pCTC#2, but in those cases, we observed only the mutated allele (T) (Figure 4). Therefore, some CTCs showed “tracking” of GBM mutations, such as promoter *TERT* mutations.

## 3. Discussion

GBM is the most aggressive malignant brain tumor in adults [22]. It has a poor prognosis [23] and current treatment options have limited effectiveness [24]. To improve outcomes for GBM patients, there is a pressing need for new therapeutic targets and noninvasive markers for disease monitoring. The recent literature has highlighted that GBM releases tumoral content that crosses the blood–brain barrier (BBB) and is detected in patients’ blood, such as CTCs [25]. However, isolating and characterizing CTCs in GBM is challenging. Several groups have reported the presence of CTCs in the blood of GBM patients using different approaches to obtain a level of detection varying from 20% to 70% [13,14,16,19]. No optimal technique exists to isolate CTCs from patients with GBM. Currently, no markers can specifically confirm GBM cell in origin. In our study, leveraging on the combination of the Parsortix and DEPArray instrumentations, we set up a new approach based on isolating CTCs from blood depending on cell size and then using GFAP (glial fibrillary acidic protein) as the main tumor astrocyte cell marker [26], with EGFR (epidermal growth factor receptor) and ki67 (proliferation marker protein), as described by Krol et al. [17]. Here, in our study, unlike other studies, the approach that we used allowed us to isolate and collect a larger number of CTCs derived from one GBM patient. We analyzed a patient who underwent surgery 14 months after the first surgery for distant recurrence and found 18 CTCs confirmed as tumor cells through CNA analysis. Moreover, 12 dposCTCs and 158 pCTCs were identified. The presence of double-positive cells (double staining for cytokeratins and CD45) is a well-known phenomenon [27,28,29,30,31,32]. It has also been shown that the presence of these cells is correlated with a worse survival rate [33]. In fact, our results showed that dposCTCs are indeed cancer cells because they carry CNAs (Figure 2 and Figure 3). One of the most widely accepted theories that can explain the double-positive cells occurrence suggests that these cells are the result of a fusion between macrophages and cancer cells [33]. To better characterize the patient’s CTCs, we also performed whole-exome sequencing of the primary tumor and the respective recurrence and compared their genetic backgrounds. We identified 210 somatic mutations in common between primary and recurrent tumors and selected three specific mutations with an allelic intratumor frequency above 30% to investigate their presence in the isolated CTCs (Figure 2 and Figure 3). The 4/6 CTCs, all the pCTCs and one dposCTC presented at least one of the selected mutations, so confirming the CTCs origin from the primary and recurrent tumors. The absence of these three mutations in some CTCs is explained by the fact that they most likely carry other somatic mutations that were not selected in our investigation. In this study, we report that the genetic background of CTCs in GBM patients presents at the chromosomal level several additional copy number alterations that are not the classical conventional GBM alterations, such as the gain and loss of chromosomes 7 and 10, respectively. This could mean that CTCs originate from subclones in which these alterations are absent, and that these are probably not correlated to cancer dissemination. Nevertheless, by sequencing specific somatic mutations, we proved that the CTCs that we collected belonged to the parental primary and recurrent tumors of origin. Moreover, to further emphasize the derivation of CTCs from recurrence, we investigated known variations in the *TERT* promoter, in addition to patient-specific mutations. In particular, the two most common somatic mutations in *TERT* promoter, C228T and C250T, which are present in almost 60% of glioma patients [34,35] were investigated in both CTCs and parental tumors. The C228T mutation was detected in tumor tissues and, in confirmation of our hypotheses, in one CTC and one pCTC. Interestingly, both CTCs showed only the T allele in position 228. This fact indicates that both cells have completely lost the wild-type allele. The loss of the wild-type allele is a phenomenon highlighted by the single-cell analysis approach, but it remains unclear when analyzing the bulk tumor. In fact, the heterozygosity of the C228T mutation that we observed at the tumor bulk level could be explained by different tumor molecular status, such as (1) one whole of single cells heterozygous for the mutation, (2) a mixture of cells homozygous for one allele or the other, and (3) both of the molecular conditions proposed above. Impressively, both CTCs analyzed showed the same loss, thus, supporting the second condition and suggesting a greater aggressiveness in disseminating behavior. Through extensive and targeted genomic characterization, we showed that CTCs have common genetic alterations with the parental tumor. However, we also showed that not all CTCs recapitulated all genetic alterations of the parental tumor, probably because of the presence of several subclonal genetic alterations in the tumor mass. For instance, the absence of *BRAF* V600E mutation in CTCs can be explained by the low frequency in the recurrence. In conclusion, while the study’s current limitation lies in the analysis of a single patient, it serves as a promising foundation for future research. Expanding the case series to include a larger cohort of GBM patients will provide a more robust understanding of CTCs’ characteristics and their clinical implications. This could potentially lead to the development of non-invasive diagnostic and monitoring tools, as well as guide the development of novel therapeutic strategies for GBM.

## 4. Materials and Methods

### 4.1. Patient and Samples

A 55-year-old man was diagnosed with a suspected brain tumor of the right temporal lobe after the occurrence of seizures. He underwent gross total resection in Livorno Hospital, and histomolecular diagnosis showed *IDH* wild-type GBM and *MGMT* unmethylated. He underwent standard radiochemotherapy and subsequent adjuvant chemotherapy with temozolomide (TMZ). Subsequently, he was treated with consecutive cycles of adjuvant TMZ. After 10 cycles, 11 months after the end of concomitant radio-chemotherapy, he developed left motor impairment and drowsiness. Progression disease was diagnosed with distant recurrence from the original site in the right frontal lobe (Figure 5). He underwent a second surgery with gross total tumor excision and blood collection for CTCs (8 mL of blood was collected in tubes containing EDTA). The blood collection was performed before tumor excision. At the time of the second surgery, our protocol for fresh GBM tumor collection was established. Surgically resected recurrent tumor was collected directly from the surgical room. The primary FFPE tumor was recovered from the archives of the Livorno Hospital, since, at that time, our fresh tissue collection protocol was not yet active.

The study was performed in accordance with the Declaration of Helsinki and the sample collection protocol was approved by the Ethics Committee of the University Hospital of Pisa (787/2015). The patient signed a written informed consent to participate in the study. Patient’s data and samples have been completely anonymized.

### 4.2. Study Design

We collected blood from the patient for CTCs enrichment and whole exome analysis (to eliminate the germinal signature). We also performed deep whole-exome analysis of the DNA extracted from the recurrent fresh tumor tissue. We performed a deep whole-exome analysis of DNA extracted from the FFPE primary tumor. The study design is illustrated in Figure 5.

### 4.3. CTCs Enrichment and Isolation

#### 4.3.1. CTCs Enrichment from Peripheral Whole Blood

CTCs were enriched using Parsortix Technology (Angle plc, Guildford, Surrey, UK) with a 6.5 µm separation cassette following the manufacturer’s instructions from blood (8 mL in EDTA tubes).

#### 4.3.2. Immunofluorescence of Single-Cell Suspensions

The cell suspensions were fixed by adding 400 µL of Paraformaldehyde 4% for 20 min. The antibodies chosen for staining were anti-GFAP APC (130-118-489, Miltenyi Biotec, Bergisch Gladbach, Germany), anti-EGFR PE (130-110-586, Miltenyi Biotec), and anti-Ki67 PE (130-120-417, Miltenyi Biotec) for CTCs; anti-CD45 FITC (130-110-631, Miltenyi Biotec) as a negative control; and Hoechst 33342 (62249, Thermo Fisher Scientific, Waltham, MA, USA) for nuclei. CD45, which was used as a negative control, is a marker for nucleated hematopoietic cells (non-tumoral cells).

#### 4.3.3. Single Cell Isolation by DEPArray^TM^ NxT

Single cells were isolated and sorted using a DEPArray NxT (Menarini, Silicon Biosystems, Bologna, Italy). Single cells were selected manually based on fluorescence labeling and morphology. The CTCs were sorted in 0.2 mL tubes: single cells or groups of single cells per tube, according to our interest.

#### 4.3.4. Ampli1™ Whole Genome Amplification and Low-Pass Analysis on CTCs

Whole-genome amplification of all recovered single cells was performed using the Ampli1™ WGA Kit version 02 (Menarini, Silicon Biosystems, Bologna, Italy), following the manufacturer’s instructions. Sequencing-ready libraries were prepared with the Ampli™ LowPass Kit (Menarini, Silicon Biosystems) to detect chromosomal aneuploidies and copy number alterations (CNAs) with a low sequencing depth. To sequence our libraries, we used the Ion 520/530-OT2 kit (Ion Torrent, Life Technologies, Grand Island, NY, USA) with the Ion 530 Chip (Ion Torrent). The runs were performed on an Ion S5 system (Ion Torrent).

#### 4.3.5. CNA Calling

The data obtained from the low-pass whole-genome sequencing were processed using the IchorCNA tool. This was optimized for low-coverage (~0.1X) sequencing. The CNA-segmented number profiles obtained from IchorCNA were processed using the CNApp tool [36] with default cutoffs.

#### 4.3.6. DNA Extraction and Whole Exome Sequencing

For bulk sequencing, genomic DNA was extracted directly from up to 50 mg of fresh recurrence tissue using the Maxwell^®^ 16 Instrument with the Maxwell^®^ 16 Tissue DNA Purification Kit (Promega, Madison, WI, USA) and from 4 × 10 μm FFPE tissue sections of GBM primary tumor. Libraries for deep whole exome sequencing were prepared using the Illumina DNA Prep with Enrichment kit (Illumina, San Diego, CA, USA) and they were run on a NextSeq 550 High Output Cartridge (300 cycles) with an average coverage of 200X. Paired-end sequencing was performed on a NextSeq 500 system (Illumina, San Diego, CA, USA) with 151 bp sequencing. DNA was also extracted from whole blood (200 µL), and whole-exome analysis was performed following the same procedure described above to obtain the germline signature.

#### 4.3.7. NGS Analysis

Raw data in fastq format were first analyzed for quality using FastQC v0.11.9 software (https://www.bioinformatics.babraham.ac.uk/projects/fastqc/ (Accessed 26 March 2021). Exome analysis was performed using the SeqMule pipeline [37] that enables the performance of all steps required for variant calling (alignment, re-alignment, quality score recalibration, and variant calling). To obtain somatic mutations, Mutect2 (https://gatk.broadinstitute.org/hc/en-us/articles/360037593851-Mutect2) (Accessed 8 April 2021) was used to pair the tumor tissue DNA with its corresponding control blood germinal DNA. Somatic variant annotation was performed using Illumina Variant Interpreter (https://variantinterpreter.informatics.illumina.com/home) (Accessed 31 March 2021). Summary tables and graphs were created using the R package, Maftools (https://bioconductor.org/packages/release/bioc/html/maftools.html) (Accessed 22 April 2021).

The Integrative Onco Genomics (IntOGEn) framework (https://intogen.org) (Accessed 8 April 2021) was used to identify the presence of driver genes in the mutated genes. Moreover, we questioned the VarSome (https://varsome.com/) (Accessed 15 April 2021) engine, which consists of a set of tools and platforms to analyze human genetic variations.

#### 4.3.8. Molecular Characterization of CTCs

To detect *TERT* promoter variations (C228T and C250T), *PKRCB* (c.430G > A), *TBX1* (c.815C > T), and *COG5* (c.1554T > G) SNPs, primers were designed using the Primer3 software. The primers used were as follows:

*PKRCB*, forward 5′-CAGCCTAAGCCACATCCCCTC-3′

*PKRCB*, reverse 5′-GTCGATGTGGGCCTGGATGTA-3′

*TBX1*, forward 5′-CCCACGCAAAGATAGCGAGA-3′

*TBX1*, reverse 5′-AGAGGCGTTGAATCCGCTC-3′

*COG5*, forward 5′-GTTTTTCCCCCGGGTGGTC-3′

*COG5*, reverse 5′-ATATGGCACTCATCTTATGGCAA-3′

*TERT*, forward 5′-GTCCTGCCCCTTCACCTTC-3′

*TERT*, reverse 5′-AGCACCTCGCGGTAGTGGC-3′.

For the detection of the V600E *BRAF* mutation, the ddPCR Mutation Detection assay specific (Bio-Rad Laboratories, Hercules, CA, USA) was used. PCR amplification for *TERT* C228T and C250T and for *PKRCB* (c.430G > A), *TBX1* (c.815C > T), and *COG5* (c.1554T > G) was performed in a volume of 25 μL according to DreamTaq DNA Polymerase (Thermo Fisher Scientific, Waltham, MA, USA) protocol and sequenced using the BigDye Terminator v3.1 Sequencing Kit (Thermo Fisher Scientific, Waltham, MA, USA) using an ABI PRISM 3130XL Genetic Analyzer (Thermo Fisher Scientific, Waltham, MA, USA). For *BRAF* V600E, we performed digital droplet PCR (ddPCR) using the conventional method on Bio-Rad QX200TM (Bio-Rad Laboratories).

## Figures and Tables

**Figure 1 ijms-24-10147-f001:**
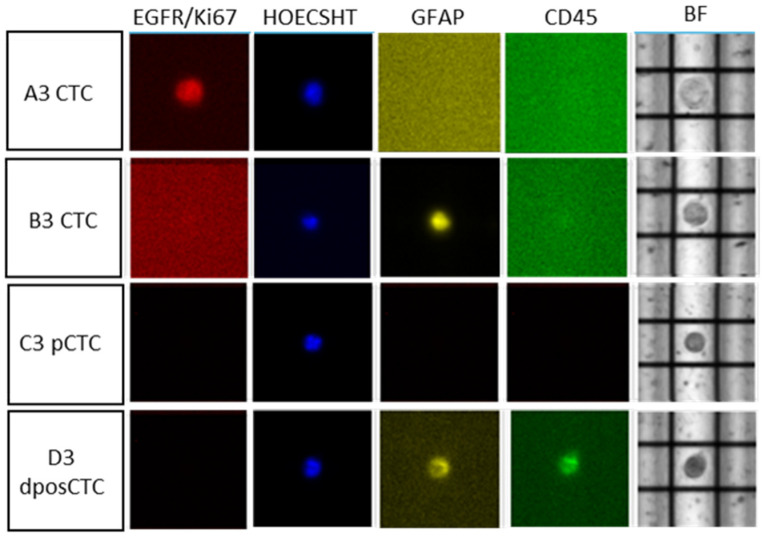
Example of DEPArray isolated CTCs from the patient’s blood (20× magnification). In red staining with EGFR/Ki67, in yellow with GFAP, in green with CD45 and in blue with Hoechst. BF: brightfield.

**Figure 2 ijms-24-10147-f002:**
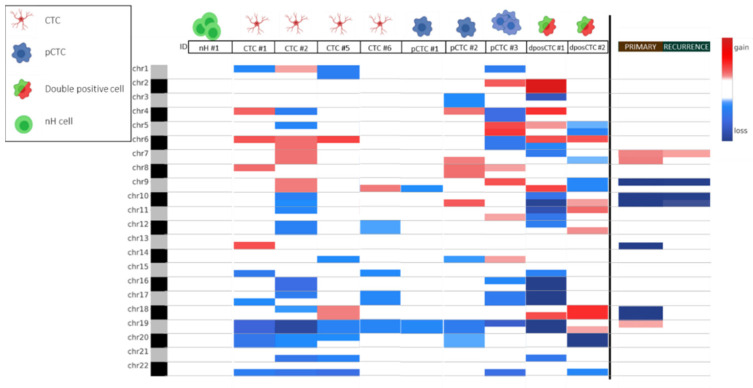
Genome-wide chromosome arm can profile heatmap for isolated CTCs obtained with CNApp tool. On the right, the CNA profiles of the primary and the recurrent tumors in bulk are shown. The chromosomal amplifications are shown in red, and the deletions in blue.

**Figure 3 ijms-24-10147-f003:**
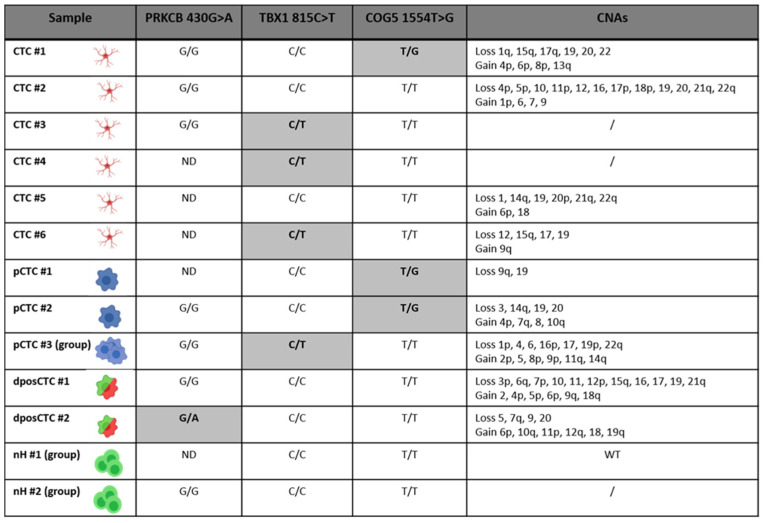
Summary of the results about the presence of the selected mutations in the isolated CTCs together with the CNA data derived from low-pass analysis *PKRCB* (c.430G > A): G/G wild type and G/A mutation; *TBX1* (c.815C > T): C/C wild type and C/T mutation; *COG5* (c.1554T > G): T/T wild type and T/G mutation. The mutated samples on grey background.

**Figure 4 ijms-24-10147-f004:**
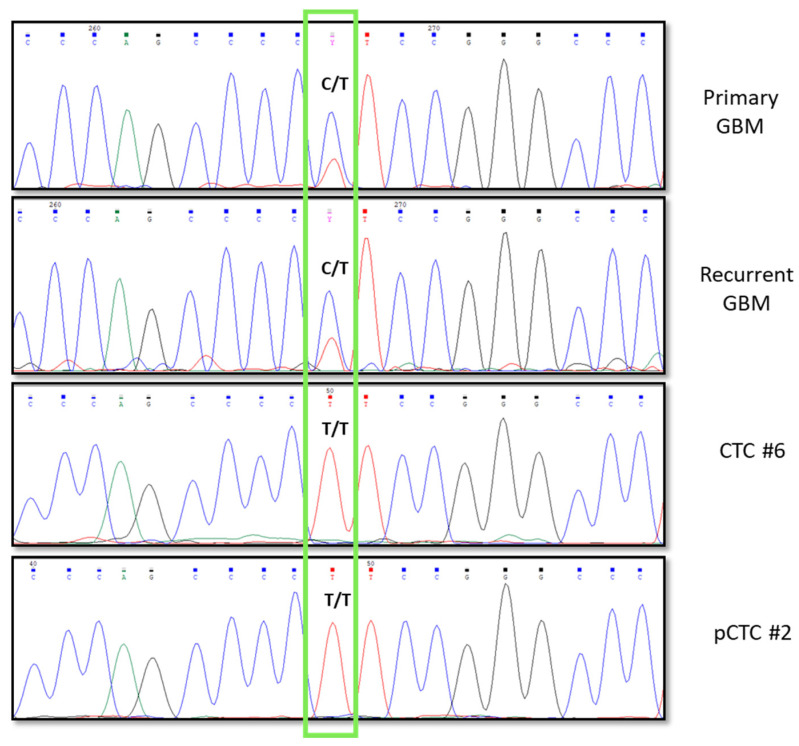
Electropherograms showing DNA sequences of C228T mutation in *TERT* promoter that was present in both the tissues (recurrence and primary GBM), and in CTC#6 and in pCTC#2. C wild type allele and T mutated allele.

**Figure 5 ijms-24-10147-f005:**
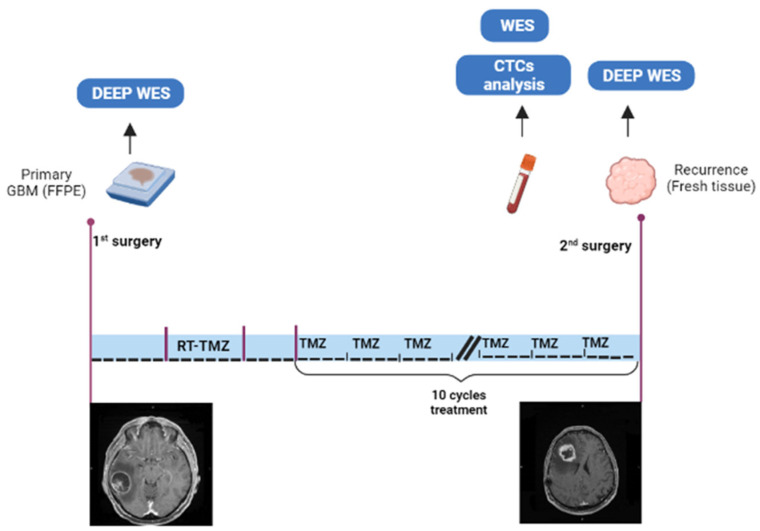
Description of the study. FFPE primary tumor collected after the first surgery, and analyzed for a deep WES (whole exome sequencing). After 14 months of follow up, after standard radio-chemotherapy and adjuvant chemotherapy, recurrence was diagnosed in the right frontal lobe. At the second surgery, blood and recurrent tumor (fresh tissue) were collected. Blood was used for CTCs isolation and WES, while the recurrence was analyzed with for a deep WES. T1 gad axial imaging from MRI before first and second surgery are shown.

## Data Availability

Data are available upon request by writing to the corresponding author.

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
