# Peer review of "Innovative Approach to Isolate and Characterize Glioblastoma Circulating Tumor Cells and Correlation with Tumor Mutational Status"

_ijms, 2023, doi:10.3390/ijms241210147_

Round 1

Reviewer 1 Report

Lessi and colleagues detected and analyzed CTCs in the blood from a glioblastoma patient at the time of recurrency, 14 months after removal of the original tumor and after radio-chemotherapy and adjuvant chemotherapy. The authors analyzed the DNA of the original FFPE tissue and the fresh tumor tissue of the recurrent tumor by deep WES. CTCs were first enriched by size, then selected for at least one marker GFAP, EGFR, or Ki67. Unstained cells (also excluding CD45) with a nucleus were also included due to their typical size and shape of CTCs. Genetic characterization by established methods included WES, CNA, and detection of selected mutations in CTCs. The differences in mutational characteristics of CTCs and the tumor tissues are discussed.  The study confirms the detectability of CTCs from blood in glioblastoma patients to generate information on the evolution of the tumor.

The manuscript can be further improved by clarifying some minor issues.

L.43: CTCs can also be derived from metastases, right?

L. 44ff (and many more in first part of manuscript): Correct all citations to numbers.

L.44: it may be mentioned somewhere that only very few CTCs (of any type of cancer) will develop into a metastatic lesion (but usually never in GBM). It might also be pointed out more than just in the abstract, that GBM only rarely metastasize hematogenously, but how surprising the detection of CTCs in GBM patients then was. A current review on liquid biopsy and glioblastoma, or liquid biopsy and primary brain tumors may be included as well.

L. 83ff: might be a little bit confusing for the reader. What does “astrocytes” or “hematopoietic” cells really mean in this context: normal astrocytes vs. CTCs. Perhaps, use of “astrocytic” markers, and “hematopoietic” (or marker for astrocytes …), or similar, may be less confusing, since CTCs are no normal astrocytes.

L.268. Please clarify or correct: According to the Figure 5, the blood collection took place “before” the 2nd surgery. Please indicate in the text (or also the figure), if the blood sample was taken before the tumor was removed (or during the removal). In case that the blood was taken during or after the surgery, correct the figure and discuss the likelihood that CTCs may have been released into the blood by this procedure – it is not very clear right now.

Perhaps, discuss in more detail: double postives Dpos – fusion with leukocytes or extracellular vesicles thereof?

Author Response

We would like to thank the reviewer for his/her thoughtful comments. We will answer point by point to the reviewer’s comments:

L.43: CTCs can also be derived from metastases, right?

L.43 CTCs are defined as cells that detach from a tumor mass and therefore theoretically can also detach from a mestastasis.

L.44ff (and many more in first part of manuscript): Correct all citations to numbers.

L.44ff We added the right numbers to the citations.

L.44: it may be mentioned somewhere that only very few CTCs (of any type of cancer) will develop into a metastatic lesion (but usually never in GBM). It might also be pointed out more than just in the abstract, that GBM only rarely metastasize hematogenously, but how surprising the detection of CTCs in GBM patients then was. A current review on liquid biopsy and glioblastoma, or liquid biopsy and primary brain tumors may be included as well.

L.44 We added from line 59 to 64 a few sentences to better explain the rarity of metastasis in GBM and the various theories. Moreover we added a recent review (L.71) in the bibliography.

L.83ff: might be a little bit confusing for the reader. What does “astrocytes” or “hematopoietic” cells really mean in this context: normal astrocytes vs. CTCs. Perhaps, use of “astrocytic” markers, and “hematopoietic” (or marker for astrocytes …), or similar, may be less confusing, since CTCs are no normal astrocytes.

L.83ff We agree with the reviewer that the use of these terms could be confusing for the reader, we changed with “tumor astrocytes” instead of “astrocytes” and we added that the nucleated hematopoietic cells are non-tumoral cells.

L.268. Please clarify or correct: According to the Figure 5, the blood collection took place “before” the 2nd surgery. Please indicate in the text (or also the figure), if the blood sample was taken before the tumor was removed (or during the removal). In case that the blood was taken during or after the surgery, correct the figure and discuss the likelihood that CTCs may have been released into the blood by this procedure – it is not very clear right now.

L.268 The blood sampling for CTCs was done at the time of surgery, as soon as the surgery starts, therefore before the tumor excision, so any cell release due to this is not to be taken into account. We added to L.271 a sentence explaining this. In the figure 5 the both the blood draw and the tumor excision appears togheter (see the second dot with written “2nd surgery”) precisely because it is done during the surgery. If it does not clear we can make an edit.

Perhaps, discuss in more detail: double postives Dpos – fusion with leukocytes or extracellular vesicles thereof?

Finally, we agree with the reviewer that we did not discuss the presence of the double-positive cells very well. They are mainly CTCs fused with leukocytes. We added to line 209 some sentences and references about it.

We would like to thank the referee again for taking the time to review our manuscript.

Reviewer 2 Report

  • A brief summary 

Lessi and colleagues present a very detailed analysis of circulating tumor cells in a patient with recurrent GBM.

  • General concept comments

The use of multiple genomic analyses to compare circulating tumor cells in the setting of recurrent GBM and comparing it to recurrent and diagnostic tissue samples creates a complete picture of the genomic landscape in CTCs.  However, this study is limited to a single patient.  Although the study is well done, the sweeping conclusions that the authors make cannot be supported by a single patient analysis.

  • Specific comments 

21-22 – this sentence creates the impression that hematogenous (not hematogenic) spread of brain tumors has not been established.  This has been well known in GBM for some time, as multiple solid organ transplant recipients developed extracranial GBM after receiving organs from GBM patients.  Agree that there is not a standard for identifying GBM CTCs as there is for epithelial cancers but many studies have been able to do so on small cohorts of patients. 

25 – “genotyped” is misspelled

34 – reference to “patients” pleural when this study only involves one patient, single

80 – the description of the CTCs is clear and the figure further emphasizes this

143 – The mention of BRAF V600E seems to come out of nowhere and there is no mention of the patient’s treatment until the discussion and then the methods.  There should be mention that targeted BRAF therapy was not used to account for the diminishing percentage of V600E mutant cells in the tumor when this is first mentioned.

204-205 – The authors mention that CTCs double positive for CD45 and cytokeratins are well known in the epithelial cancer literature.  Since GBM is not an epithelial tumor, as the authors point out earlier, this phenomenon has not been as robustly described in brain cancer.  It seems to be a missed opportunity to have investigated these cells further.

Some minor wording and spelling

Author Response

We would like to thank the reviewer for his/her thoughtful comments. We will answer point by point to the reviewer’s comments:

  • 21-22 – this sentence creates the impression that hematogenous (not hematogenic) spread of brain tumors has not been established.This has been well known in GBM for some time, as multiple solid organ transplant recipients developed extracranial GBM after receiving organs from GBM patients.  Agree that there is not a standard for identifying GBM CTCs as there is for epithelial cancers but many studies have been able to do so on small cohorts of patients.

As you suggested we changed the sentence L21-22 and we added a comment to L63 to better underline your point

  • 25 – “genotyped” is misspelled

We corrected the error

  • 34 – reference to “patients” pleural when this study only involves one patient, single

We corrected the error

  • 80 – the description of the CTCs is clear and the figure further emphasizes this

Thank you for your comment

  • 143 – The mention of BRAF V600E seems to come out of nowhere and there is no mention of the patient’s treatment until the discussion and then the methods.There should be mention that targeted BRAF therapy was not used to account for the diminishing percentage of V600E mutant cells in the tumor when this is first mentioned.

As you suggested we added some sentences to better explain BRAF V600E mutation analysis in our sample. We added something at line 83-152 and 256.

  • 204-205 – The authors mention that CTCs double positive for CD45 and cytokeratins are well known in the epithelial cancer literature.Since GBM is not an epithelial tumor, as the authors point out earlier, this phenomenon has not been as robustly described in brain cancer.  It seems to be a missed opportunity to have investigated these cells further

As you suggested we underlined better the double positive cells presence with a sentence at line 215-218

We would like to thank the referee again for taking the time to review our manuscript.

Reviewer 3 Report

In this manuscript by Francesca Lessi, et al., authors defined approaches to isolate CTCs from blood of patients with recurrent glioblastoma and characterize glioblastoma circulating tumor cells. Also corelated their mutational signature to primary GBM tissues available from archives.

Overall the manuscript is well written and worth considered for the publication however some improvements needed for the improvement.

Specific comments: 

1) The manuscript reads like a review than a research article, some formatting changes needed in introduction and discussion

2) Discussion should be compressed, and the importance of current work needed to be highlighted.

3) The sample number of CTCs (9/13) looks very less, can authors use database information from other similar samples to confirm the mutational signature.

Author Response

We would like to thank the reviewer for his/her thoughtful comments. We will answer point by point to the reviewer’s comments:

  • The manuscript reads like a review than a research article, some formatting changes needed in introduction and discussion

As you suggested some changes have been made in Introduction and Discussion Sections, in particular at line 59-66; 73-75 and 83 in the Introduction and line 215-219 and 256-266 in the Discussion.

  • Discussion should be compressed, and the importance of current work needed to be highlighted.

We changed the last part of the Discussion (L.256-266) to highlight the importance of our work as you suggest

  • The sample number of CTCs (9/13) looks very less, can authors use database information from other similar samples to confirm the mutational signature.

Thank  you for your comment but unfortunately a database to confirm the mutational signature of CTCs doesn’t exist for GBM patients.

We would like to thank the referee again for taking the time to review our manuscript.